# Amorphous Blue Phase III: Structure, Materials, and Properties

**DOI:** 10.3390/ma17061291

**Published:** 2024-03-11

**Authors:** Atsushi Yoshizawa

**Affiliations:** Department of Frontier Materials Chemistry, Graduate School of Science and Technology, Hirosaki University, 3 Bunkyo-cho, Hirosaki 036-8561, Aomori, Japan; ayoshiza@hirosaki-u.ac.jp

**Keywords:** liquid crystals, blue phase, chirality, electro-optical effect, display

## Abstract

Blue phases (BPs) have a frustrated structure stabilized by chirality-dependent defects. They are classified into three categories: blue phase I (BPI), blue phase II (BPII), and blue phase III (BPIII). Among them, BPIII has recently attracted much attention due to its elusive amorphous structure and high-contrast electro-optical response. However, its structure has remained unelucidated, and the molecular design for stabilizing BPIII is still unclear. We present the following findings in this review. (1) BPIII is a spaghetti-like tangled arrangement of double-twist cylinders with characteristic dynamics. (2) Molecular biaxiality and flexibility contribute to stabilize BPIII. (3) BPIII exhibits submillisecond response, high contrast, and wide-viewing angle at room temperature without surface treatment or an optical compensation film. It was free from both hysteresis and residual transmittance. The electro-optical effects are explained in relation to the revealed structure of BPIII. Finally, we discuss the memory effect of a polymer network derived from the defects of BPIII.

## 1. Introduction

Blue phases are of particular interest because they have a frustrated structure that is stabilized by lattice defects [1,2,3,4,5,6,7,8,9,10,11,12]. Generally, they appear in a very narrow temperature range (less than 2 K). Blue phases consist of double-twist cylinders (DTCs). Molecules are twisted from –45° at one end to 45° at the other end within each cylinder. They are classified into three categories depending on the cylinders’ packing structure: blue phase I (BPI), blue phase II (BPII), and blue phase III (BPIII) (Figure 1) [6]. BPI and BPII have a body-centered cubic structure and a simple cubic one, respectively [2,13]. The lattice parameter for BPI is one helical pitch, whereas that for BPII is a half-pitch. On the other hand, BPIII is thought to have an amorphous structure.

Cubic blue phases with a three-dimensional structure have recently attracted much attention due to their stimulus responsiveness, i.e., electrical, photonic, mechanical, and chemical [12]. They can be used not only for electro-optical applications but also for chemical sensors. For example, cubic BPs that change their reflection wavelength in response to chemical stimulus can give the basis of calorimetric sensors. BP-based calorimetric sensors to detect humidity [15] or toluene [16] have been reported. Furthermore, the BP structure has the potential to perform as a template to produce a higher-ordered structure. Blue phase liquid crystals are expected to create functional soft materials [12,17].

The structure of BPIII has remained unresolved. A number of models for BPIII have been proposed [18,19,20,21,22]. Among them, the double-twist model [19] and cubic domain model [20] are realistic. Recently, Henrich et al. proposed that an amorphous disclination network in BPIII is locally close to cubic BPII [23]. On the other hand, the more recent simulation by Paul and Saha supports the theoretical proposition of the spaghetti-like arrangement of the double-twist cylinders [24]. In addition to the simulation studies, some experimental studies have been conducted to identify the structure of BPIII.

A lot of research has been performed to widen the blue phase temperature range [9,10]. The formation of DTCs inevitably accompanies line defects because the constituent molecules cannot fill the space uniformly. The existence of the line defects increases the free energy of the system. Many attempts have been made to stabilize BPs by forming a defect-free system or by stabilizing double-twist structures. There are two pioneer works on obtaining BP with a wide temperature range. Kikuchi et al. reported polymer-stabilized BPI in which the temperature range was extended to 60 K [25]. With respect to the molecular design of BPs, Coles and Pivnenko reported that a nematic mixture of bimesogenic compounds doped with a small quantity of a chiral additive showed BPI of an extremely wide temperature range (40 K) [26]. In addition, hydrogen-bonded assemblies for stabilizing cubic BPs were reported [27,28]. Although many BP materials have been developed, a clear structure–property relationship in molecular design for BPIII has not been still established.

Blue phases are recognized as offering the potential for application as fast light modulators, tunable photonic crystals, or next-generation displays. BPI and BPII consisting of three-dimensional periodic structures are potentially useful for application as tunable photonic crystals [29,30,31]. Fast tunable reflection was also observed in amorphous BPIII [32]. For display applications, blue phases are optically isotropic; therefore, they exhibit a black state without surface treatment such as an alignment layer. The response time is in the submillisecond range [33,34]. A blue phase liquid crystal display (BPLCD) mode using BPI was developed in earlier studies [35]; however, some critical problems such as hysteresis in voltage–transmittance curves [36] and residual transmittance have emerged. This is partially due to the three-dimensional cubic structure. On the other hand, BIII is thought to be amorphous without a 3D structure. We reported high-contrast and hysteresis-free electro-optical response in BPIII without surface treatment [37,38]. The origin of the hysteresis-free switching is not clear yet.

We clarify the structure of BPIII based on recent theoretical studies and our experimental results in this review. The understanding of BPIII structure gives useful information to discuss the structure–property relationships in BPIII. We propose a rational molecular design for stabilizing BPIII and explain the characteristic electro-optical effects in relation to the revealed structure of BPIII. Finally, we discuss the mechanism of the memory effect of a polymer network derived from the defects of BPIII. The presented findings provide important information to create BPIII-based functional materials.

## 2. Phase Structure of BPIII

First of all, let us show the textures of blue phases. Figure 2a,b are optical textures of BPI and BPII under crossed polarizers, respectively. The platelet textures are characteristic of cubic BPs. Both BPI and BPII have cubic structures with lattice parameters of approximately visible wavelengths. Therefore, several selective reflection wavelengths correspond to various crystal planes. Figure 2c is a foggy texture of BPIII. The cubic BPs were actively investigated by several experimental methods, such as the Kossel diagram technique [39,40], and are well understood. By contrast, the structure of BPIII remains to be solved.

Stegemeyer and Bergmann clearly identify BPIII [1,4]. In earlier studies, the experimental investigation of BPIII encountered some difficulties due to its very narrow temperature range and the limitation of compounds exhibiting BPIII. Despite this, the following important results were obtained [4,6].

(1)BPIII is the highest-temperature blue phase.(2)The visual appearance of BPIII is foggy, and it is often bluish or grayish in color. BPIII appears to be closer in structure to the isotropic liquid.(3)BPIII reflects circularly polarized light and exhibits only one broad peak [41,42].(4)A heat capacity peak between BPIII and the isotropic liquid was much larger than that between BPIII and BPII. Therefore, BPIII appears to be closer in structure to BPII than to the isotropic phase, in contrast to the visual appearance [43].(5)BPIII shows higher shear elasticity and viscosity than the BPII and isotropic phases [44].(6)Transmission electron microscopy (TEM) of a BPIII compound (**CE4**) shows that the BPIII is a spaghetti-like tangle of DTCs of diameters a fraction of the cholesteric pitch. No long-range structure or periodicity is present, although the DTCs appear aligned over distances of a few microns [45].

The TEM measurements reported by Zasadzinski et al. [45] give us useful information on the BPIII structure. Figure 3 shows TEM images of a fracture surface of **CE4** in the cholesteric, BPI and BPIII phases. Molecular structure and transition temperatures of **CE4** are also shown.

Let us describe the findings [45]. A one-dimensional periodic structure is observed in the Ch phase (Figure 3a). Each light-line–dark-line pair corresponds to one half-pitch of the helix. The full pitch length is about 240 nm. The small arrows in Figure 3b point to square patterns corresponding to fracture along a {100} plane of the cubic lattice in the BPI. A regular zigzag pattern at the large arrow is also seen. These areas show a regular periodicity in two directions in the BPI. The surface in the BPIII is covered by filamentary-shaped objects of varying diameter between 10 and 50 nm packed in a random fashion (Figure 3c). Small groups of filaments extend for 0.1–1.0 µm in a given direction before ending or changing direction. These cylindrically shaped objects are arranged in a random and liquid-like fashion.

A number of models for BPIII, i.e., emulsion model [18], double-twist model [19], cubic domain model [20], and icosahedral model [21], were presented. Among them, we think that the double twist model and the cubic domain model are realistic. The double-twist model suggests that BPIII is a spaghetti-like tangle of DTCs. On the other hand, the cubic domain model includes a possibility that BPIII retains local cubic structure, but only over short correlated regions.

The structure of BPIII has remained unresolved, despite much effort. Recently, Henrich et al. reported that BPIII is an amorphous network of disclination lines, which is thermodynamically and kinetically stabilized over cubic BPs at intermediate chiralities by using large-scale simulations of the blue phases of cholesteric liquid crystals [23]. Aperiodic structures are generated by placing a localized nucleus of BPI or BPII in a cholesteric matrix. For low chirality, the initial defects washed out to leave a cholesteric phase. For a large regime of intermediate chiralities, the dilute double-twisted regions grow and rearrange dramatically to form a whole network of disclinations, which very slowly creeps to an amorphous end state (Figure 4). The authors propose that an amorphous disclination network is locally close to BPII.

More recently, two interesting simulation studies on BPIII have been reported. The study reported by Paul and Saha supports the theoretical proposition of the spaghetti-like arrangement of the double-twist cylinders as a model of BPIII [24]. Pišljar and Muševič et al. reproduced the bulk BPIII of mutually enmeshed structures of singular defect lines (yellow) and skyrmion filament-DTCs (Figure 5) [46]. They concluded as follows. BPIII is a topologically protected liquid of strongly fluctuating skyrmion filaments with special dynamics. The basic building units of BPIII are the skyrmion filaments. They form highly dynamic, topologically charged entities, which cannot be annihilated due to the elastic energy barrier.

Experimental investigations on the structure of BPIII have been carried out. One of them included observation of a polymer network derived from the defects of BPIII. It was based on the method reported by Coles et al., as shown below. They fabricated a polymer template that prints the defect structures of BPI, as portrayed in Figure 6 [47]. A mixture of chiral LCs and reactive mesogens form a BPI phase (Figure 6a). The reactive mesogens were photopolymerized in the BPI (Figure 6b). The template was obtained by removing the low-molecular-mass materials (Figure 6c). By refilling the template with an achiral nematic liquid crystal, a texture characteristic of cubic BP appeared with wide temperature ranges (Figure 6d). The polymer template shows an interesting memory effect. The effect will be discussed later.

Based on the assumption that the thus-obtained polymer network prints defect structures in the original blue phase, the morphologies of polymer networks derived from cubic BPs and BPIII have been investigated by field-emission scanning electron microscopy (FE-SEM) [48,49]. Gandhi and Chin et al. reported the fabrication of a porous polymer network that mimics the 3D structure of BPIII at the nanoscale by imprinting a reactive mesogen polymer network along topological defects in BPIII [49]. The BP mixture used for producing the polymer network comprised 4.41 wt% of a chiral dopant and 19.88 wt% of a reactive mesogen dissolved in a commercial nematic mixture. Figure 7a,b shows the confocal laser scanning microscopy (CLSM) and SEM images of the network from BPIII, respectively. The authors noted that the SEM image of the network derived from BPIII is in agreement with the simulated structure of the amorphous disclination network in BPIII reported by Henrich et al. [23].

Recently, we reported SEM images of polymer networks derived from cubic BP and BPIII [50]. We used a BP mixture of a T-shaped liquid crystal **CBDFBA** [44] (49.3 wt%), a chiral dopant **S-811** (40.7 wt%), a reactive bifunctional monomer **RM257** (7.3 wt%), a monofunctional monomer **C12A** (2.4 wt%), and a photo-initiator **DMAP** (0.3 wt%). Their molecular structures are shown in Figure 8. The phase transition temperatures of the BP mixture on cooling were Iso 65.7 °C BPIII 60.7 °C cubic BP 51.6 °C N*. We obtained a single chiral mixture exhibiting both BPIII and cubic BP with relatively wide temperature ranges suitable for polymer stabilization. According to the phase sequence of the mixture of **CBDFBA** and **S-811** [51], the cubic BP is thought to be BPII. Polymer stabilization was performed in each BP. The optical texture of the PS-cubic BP also supports that the cubic BP is BPII. After removing low-molar-mass materials, the corresponding polymer networks were obtained.

Figure 9 shows the FE-SEM images of the BPII and BPIII networks. Marked differences can be seen in the 3D structure between them. The pore sizes, defined in diameters, are 170 ± 70 nm and 180 ± 70 nm in the BPII and BPIII networks, respectively. They correspond to a diameter of double-twisted cylinders in each BP. On the other hand, the polymer chain widths are about 80 nm and 130 ± 50 nm in the BPII and BPIII polymer networks, respectively. The BPIII polymer network was thicker than the cubic one. Many reactive monomers accumulate in the defect regions in BPIII, reflecting that the defect structure of BPIII is disordered with fluidity. Therefore, the DTCs in BPIII easily rearrange in a liquid-like fashion. The FE-SEM observation indicates that the DTCs in the BPII are regularly packed to form the 3D periodical structure, while those in the BPIII are arranged in a random and tangled fashion. The entropy-driven tangled arrangement of the DTCs might produce higher viscosity in BPIII. BPIII has characteristic dynamic behavior, and the presented SEM image shows a time-average structure of BPIII. The FE-SEM study supports not a cubic domain model, but a spaghetti-like double-twist model. Recent simulations [24,46] and our FE-SEM measurements [50] reveal that BPIII is a spaghetti-like tangled arrangement of DTCs with characteristic dynamics.

## 3. Materials Stabilizing BPIII

There are two approaches to widening the BP temperature range, as described in Section 1. The first is forming a defect-free state using polymers [25,34,52] or nanoparticles [53]. The other approach is molecular design of BP LCs stabilizing DTC structures. Blue phases consisting of DTCs inevitably accompany defects. If the free energy increase caused by the defect formation is compensated, BPs with a wide temperature range can be obtained. Kitzerow et al. produced polymer networks exhibiting a blue phase structure at room temperature by in situ photopolymerization of mixtures containing a chiral and a non-chiral diacrylate [54]. Kikuchi et al. performed polymer stabilization using a reactive mesogenic monomer [25]. Polymer stabilization has been applied not only to cubic BPI but also to amorphous BPIII [14,49,55,56]. Let us introduce our results here [14]. We used a BPIII mixture containing a small amount of dimeric LC with an even-numbered methylene spacer (**8PY8OCB**) for the polymer stabilization. The components of the sample were 64.2 wt% of a nematic mixture, 3.4 wt% of **8PY8OCB**, 11.9 wt% of a chiral compound **ISO-(6OBA)_2_**, 20 wt% of a binary monomer mixture (**RM257** (50 wt%) and **C12A** (50 wt%)), and 0.5 wt% of **DMAP**. The nematic mixture consists of 4-pentyl-4′- cyanobiphenyl (50 wt%), 4-octyloxy-4′-cyanobiphenyl (35 wt%) and 4-pentyl-4′’-cyanoterphenyl (15 wt%). Molecular structures of the dimeric LC and **ISO-(6OBA)_2_** are shown in Figure 10. The phase transition temperatures of the BP mixture were Iso 75.2 °C BPIII 71.5 °C N* and those of the sample after photopolymerization were Iso 80 °C BPIII. No further phase transition was observed until 0 °C. Doping a small amount of **8PY8OCB** to the low-molecular-weight blue phase LC increases the Iso–BPIII transition temperature. The PS-BPIII showed electro-optical behavior characteristic to BPIII.

Nanoparticles have been used for stabilizing BPIII. CdSe nanoparticles, where the surface was treated with oleyl amine and trioctyl phosphine, were mixed with a chiral LC: (*S*)-4-(2-methylbutyl)phenyl 4′-octylbiphenyl-4-carboxylate (**CE8**). The mixture exhibited a phase sequence of Iso–BPIII–BPII–BPI-N*–SmA* [57]. The temperature vs. composition phase diagram is shown in Figure 11. The Iso–BPIII transition temperature remains almost the same as that of **CE8** for all the studied mixtures. The BPIII–BPI and BPI–N* transition temperatures decrease markedly with increasing *x*. The temperature range of BPIII is 20 K at *x* = 0.20. The aggregation of nanoparticles at disclination lines might stabilize BP III.

As described in Section 2, Gandhi and Chin et al. applied the BP-polymer templating process reported by Castles et al. [47] to obtain a polymer network that mimics the 3D structure of BPIII [49]. The polymer network refilled with 4-cyano-4′-pentylbiphenyl (**5CB**) exhibited a texture and reflection spectrum that is very similar to the original PS-BPIII. They compared the electrooptical behavior of the refilled network with that of the PS-BPIII. No significant difference exists between them in the EO properties, which will be described in more detail in Section 4.

Polymer stabilization and BP-polymer templating are interesting and powerful methods to obtain BPIII with a wide temperature range. However, from a practical viewpoint, they are much more complicated to apply to a conventional fabrication process used for nematic LCDs. The development of BPIII materials consisting of low-molecular-mass compounds is necessary for the application to display devices. However, the structure–property relationship in BPIII is not clear; therefore, we do not have a molecular design concept for BPIII. With respect to nematic and smectic phases, anisotropic intermolecular interactions, i.e., hard-core repulsion and electrostatic interactions, are responsible for the stabilization of the thermotropic liquid-crystalline phases. Chemists design LC molecules with these factors. We easily obtain a chiral nematic LC by doping a chiral compound to a nematic liquid crystal. Unfortunately, we do not have a simple method for transforming an achiral LC phase to a chiral BP.

In a low-molecular-mass system, amorphous BPIII was firstly stabilized by a chiral T-shaped compound (**T-1**) [58] (Figure 12). Compound **T-1** exhibited BPIII with a temperature range of 13 K on cooling. Sato and Yoshizawa introduced a polar group into a chiral T-shaped system to couple the molecule with an electric field and subsequently prepared compound **T-2** possessing a terminal cyano group exhibiting BPIII and N* phased on cooling (Figure 12). It showed high-contrast electro-optical switching in BPIII [37]. T-shaped compounds stabilize BPIII, and the effect of BPIII stabilization is explained in terms of molecular biaxiality [9,58]. A dielectric study of **T-2** in Iso and BPIII revealed that two different modes of rotation around the short axis can exist [59], supporting the inference that the molecular biaxiality stabilizes DTCs in the BPIII. On the other hand, a U-shaped binaphthyl derivative **U-1** exhibited a cubic BP with 32 K [60] (Figure 12). Although both T-shaped and U-shaped compounds have a molecular biaxiality, the more flexible T-shaped compounds stabilize BPIII.

Furthermore, some H-shaped compounds induced BPIII in their mixtures doped with 15 wt% of **ISO-(6OBA)_2_** [61]. Their molecular structures and transition temperatures are shown in Figure 13. Mixture A containing **H-1** exhibited a cooling phase sequence of Iso 89.6 °C BPIII 75.5 °C N* 16.8 °C glass. Mixture B containing **H-2** exhibited that of Iso 36.7 °C BPIII 5.5 °C glass. For comparison between the oligomers and their corresponding monomer, we doped 15 wt% of the chiral compound into the monomeric compound, 4-(4-cyanophenyl)phenyl 4-octyloxybenzoate (**CPOB**) (Figure 13). The chiral mixture (Mixture C) shows a phase sequence of Iso 202.4 °C cubic BP 195.5 °C N* 49.2 °C Cry on cooling. The optical textures of BPIII for Mixture B and cubic BP for Mixture C are shown in Figure 14. Compounds **H-1** and **H-2** with flexibility induce much wider BP temperature ranges than the monomeric compound **CPOB**. Particularly, they stabilize amorphous BPIII. The flexibility plays an important role in stabilizing BPIII. We surmise that the chiral dopant induces a transiently twisted conformation of each H-shaped compound in the BPIII, as shown in Figure 15.

The structure–property relationship in the T-shaped and H-shaped compounds reveals that molecular biaxiality and flexibility are important factors in the appearance of BPIII with DTCs in a random and tangle fashion. The molecular biaxiality stabilizes the DTCs. The disorder of a flexible molecule itself might induce the entropy-driven tangled arrangement of the DTCs in amorphous BPIII.

Yelamaggad et al. reported liquid crystal dimers **USD-1-*n*** possessing bent-core LC and cholesterol units [62] (Figure 16). The liquid-crystalline properties of the dimers exhibit pronounced dependence on the parity of the flexible spacer. **USD-1-3**, **USD-1-5**, and **USD-1-7** of even parity (the sum of n and C of CO) exhibited BPIII with a wide temperature range. However, **USD-1-4** of odd parity showed an Iso–Col transition.

Taushanoff et al. reported achiral nematic bent-core liquid crystals (BCN LCs) stabilizing BPIII selectively (Figure 17) [63]. BPIII was induced by doping a chiral compound **ISO-(6OBA)_2_** to each BCN LC. When **ClPbis10BB** was doped with a small amount of the chiral compound, and the BPIII replaced their original nematic phase (Figure 18). Electro-optical switching was observed for the bent-core system in BPIII.

The authors explained the stabilization of the BPIII in BCN LCs as follows. The bent-core LC **ClPbis10BB** exhibits the presence of smectic clusters in the whole nematic region. Nanosized smectic clusters interrupt the long-range development of the BPI or BPII lattice structures and randomize their evolution, resulting in an amorphous structure, as shown in Figure 19.

Chiang and Lin et al. synthesized two chiral 1,3,4-oxadiazole-based bent-core LCs (**OXD7*** and **OXD5B7F***) that display BPIII (34 K and 7 K, respectively), as shown in Figure 20 [64]. The authors noted that the HTP and the molecular biaxiality are important factors for stabilizing BPIII.

Recently, the twist–bend nematic (N_TB_) phase showing domains with opposite handedness was observed for members of achiral α,β-bis-4-(40-cyanobiphenyl)alkanes with flexible odd-numbered methylene spacers [65]. When a flexible odd-membered dimer exhibiting the N_TB_ phase was doped with a chiral compound (Figure 21), BPIII was induced due to the large destabilization of the nematic phase [66]. This is consistent with the results observed for the bent-core LC that did not exhibit the N_TB_ phase doped with **ISO-(6OBA)_2_** (Figure 18). The thermal stability of the N_TB_ phase remains almost constant as a function of dopant concentration. It is not clear how the N_TB_ phase affects the stabilization of BPIII.

Recently, Cordoyiannis et al. demonstrated that quantum dots essentially stabilize BPIII [67]. Khatun and Nari et al. reported stabilizing BPIII by using a twist–bend nematic LC and surface-functionalized quantum dots (QDs: CdSe (core)/ZnS (shell) tethered with octadecylamine) [68]. Adding a twist–bend nematic dimer **N_TB_ LC** (Figure 22) to a mixture comprising a nematic liquid crystal mixture and a chiral dopant **S-811** stabilizes the cubic BPI, as shown in Figure 23a. On the other hand, the mixture with 20 wt% of **Bent-core LC** (Figure 22) exhibited BPI with a thermal range of about 3.5 K. The **N_TB_ LC** is more effective in enhancing the thermal range of cubic BPI than the **Bent-core LC**. This is due to the ultralow bend elastic constant and saddle-splay deformation inherent to the **N_TB_ LC**. Doping of a small amount (0.005 wt%) of QDs reduces the free energy associated with the topological defects, leading to a complete transformation of the cubic blue phase to an amorphous BPIII (Figure 23b). The effect of QDs depends on the concentration of **S-811**. In the case of a low concentration of **S-811**, e.g., 30–40 wt%, the addition of QDs has no effect on the thermal range of BPIII.

T-shaped LCs, H-shaped LCs, or bent-core LCs can be a host material that is transformed to BPIII LC by adding a chiral dopant. However, they increase the viscosity in their BPIII to result in slow electro-optical switching. We proposed a ternary system consisting of a rod-like nematic mixture, a BPIII stabilizer, and a chiral dopant [9,38]. The design concept is shown in Figure 24. Physical properties such as temperature range, dielectric anisotropy, and viscosity are adjusted for the formation of the nematic mixture. The BP stabilizer has biaxiality to stabilize DTCs and flexibility to induce disorder in the system. A rod-like nematic LC mixture doped with such a BPIII stabilizer might be a host LC, which is converted to BPIII LC by adding a chiral dopant with high twisting power.

## 4. Physical Properties in BPIII

Compared to traditional nematic liquid crystal displays, BPLCDs have the following advantages [34,69]: (1) they require no surface alignment; (2) their response time is in the submillisecond range; (3) their viewing angle is wide and symmetric; and (4) they have cell-gap insensitivity. On the other hand, high operating voltage, hysteresis [36], and residual transmittance have been pointed out. Figure 25 shows hysteresis and residual birefringence in a voltage–transmittance curve. The hysteresis affects the accuracy of grayscale control. Two major approaches to improve device configuration have been undertaken: (1) vertical field switching (VFS) [70] and (2) fringe field switching (FFS) [71]. Both methods also decrease the driving voltage. A low voltage (<10 V) and submillisecond response PS-BP device with VFS has been demonstrated [70].

Electric field effects on BPIII have been reported in few studies [5,72,73]. Yang and Crooker investigated a temperature–electric field phase diagram in a BP mixture showing BPIII. The BPIII became a new phase (BPE) at *E* > 2 V µm^−1^ [73]. We demonstrated the high contrast electro-optical switching in BPIII due to electric-field-induced phase transition from BPIII to N for the first time [37]. A well black state is obtainable in BPIII without surface treatment, and a homogeneous bright state is achieved in the electric-field-induced N phase [37,74]. Figure 26 presents a schematic representation of the electric-field-induced phase transition between BPIII and N. BPIII has DTCs existing randomly and appears to be macroscopically isotropic. The applied electric field transforms the double-twisted molecular organization to the anisotropic nematic orientation. Once the electric field is removed, the molecules return to form the double-twist organization in BPIII.

According to the design concept shown in Figure 24, we prepared a ternary system consisting of a conventional nematic mixture, a T-shaped BP stabilizer (Figure 27), and a chiral dopant with large HTP [38]. We prepared a host liquid crystal consisting of a nematic liquid-crystalline mixture E7 (95 mol%) and a non-chiral T-shaped compound (5 mol%). The host LC was doped with 15 wt% of a chiral dopant **ISO-(6OBA)_2_**.

Figure 28 portrays the electric-field dependence of transmittances for the forward and backward processes in the BPIII at 26 °C. Transmittance without an electric field was 0%. It was 89% with an applied field of 14 V·µm^−1^. The backward curve was the same as the forward curve. It was free from both hysteresis and residual transmittance. The defect structure in BPIII is disorder with fluidity, and the DTCs easily rearrange in a liquid-like fashion (please see Section 2). Therefore, BPIII has no original state. On the other hand, the 3D lattice structure in a cubic BP cannot fully relax to its original position in the backward process. The respective rise and decay times with an AC field of 14 V·µm^−1^ at 50 Hz were 0.4 ms and 0.8 ms. Figure 29 portrays front views (Figure 29a) and oblique views with a viewing angle of about 45° (Figure 29b) of the evaluation BPIII cell sandwiched with cross-polarized films. The BPIII cell has wide and symmetrical viewing angles because of its optical isotropy of BPIII. The electro-optical switching can be seen in Appendix A.

We compared the electro-optical properties in PS-BPIII with those in PS-cubic BP [55]. According to the design concept of a BP host LC, we prepared a host nematic mixture consisting of a rod-like nematic mixture (**5CB** (50 wt%), **8OCB** (35 wt%), and **5CT** (15 wt%)) and a BP stabilizer (10 wt%). The molecular structure of the BP stabilizer is shown in Figure 30. The host LC doped with a chiral compound **ISO-(6OBA)_2_** (10 wt%) and that with **ISO-(6OBA)_2_** (15 wt%) were for cubic BP and BPIII, respectively. Then, each BP was polymer-stabilized to give the corresponding PS-cubic BP and PS-BPIII. Figure 31 shows an optical texture of a sample in the PS-cubic BP without an electric field in the forward process (Figure 31a), that with an electric field of 14 V·µm^−1^ (Figure 31b), and that without an electric field in the backward process (Figure 31c). Polymer stabilization induced higher residual birefringence in the PS-cubic BP, as shown in Figure 31c. With respect to the corresponding PS-BPIII, the transmittance without an AC field was 0% (Figure 32a). It increased with the increase in the electric field and reached a saturation value of 84.0% at 14 V·µm^−1^ (Figure 32b). It showed no residual transmittance (Figure 32c). The PS-BPIII showed higher stability against the applied electric field than the PS-cubic BP.

Electro-optical switching in BPIII by using polymer-stabilized materials [14,49,75,76] or a low-molecular-mass system [68,77] has been reported. Gandhi and Chien et al. reported that a polymer network based on BPIII can memorize the 3D structure on the original PS-BPIII from which they are nanoengineered, and can transfer this structural information to achiral nematic LCs [49]. The electro-optical behavior of the PS-BPIII and the refilled scaffold were compared. The comparison reveals that: (1) the driving electric field is about 40 V·µm^−1^ for both PS-BPIII and the refilled scaffold, (2) the transmittance–electric field curves of the PS-BPIII and the scaffold after refilling 5CB are almost overlapping, and (3) both switch-on and switch-off times of PS-BPIII are shorter than 0.2 ms and remain in the submillisecond range after refilling the network with **5CB**. These findings demonstrate the EO-memory effects of the polymer network.

Recently, Yang et al. reported an ultrastable cubic BP from molecular synergistic self-assembly of low-molecular-mass liquid crystals [78]. In their work, a series of tailored uniaxial rod-like mesogens disfavoring the formation of blue phases are introduced into a blue phase system comprising biaxial dimeric mesogens. BP temperature range and driving voltage of the mixtures markedly depend on their compositions. Let us show one of the samples below. The compositions are presented in Figure 33. The sample consists of **BPFOn** (39.3 wt%), **TPFOn** (21.8 wt%), **TTPF** (14.1 wt%), **TP2FTF** (18.8 wt%), **R-5011** (2.0 wt%), and **ISO-(6OBA)_2_** (4.0 wt%). It showed: Iso 83.5 °C cubic BP –38.8 °C unidentified phase X. The BP temperature rage was 122.3 K. The driving voltage was about 20 Vrms at 20 °C. Increasing the concentration of fluorinated compound **TP2FTF** possessing large dielectric anisotropy (De > 0) markedly decreases the driving voltage. Though this is an approach for obtaining cubic BP materials, the findings show a possibility of a practical BPIII material being realized by using a mixture consisting of designed low-molecular-mass liquid crystals.

## 5. The Memory Effect of a Polymer Network Derived from the Defects of BPIII

As described in Section 2 and Section 4, BP polymer networks show interesting memory effects [47,49]. Let us discuss how a polymer network derived from PS-BPIII can memorize the structure of BPIII and why the polymer network can transform a conventional nematic LC to BPIII. We investigated the polymer network derived from BPIII (see Section 2) by using circular dichroism spectroscopy (CD) [50]. Figure 34 shows CD spectra of the BPIII polymer network from the BPIII mixture containing **S-811** (blue) or **R-811** (red). Both networks exhibited a strong CD signal at about 350 nm, reflecting the supramolecular chirality, and their optical senses were opposite. These senses are the same as those of the corresponding BPIII. Some signals appear below 300 nm, showing mirror images, indicating that the mesogenic core of achiral monomer **RM257** occurs to twist.

Figure 35 shows a schematic model of the polymer network. We can say that the chiral field originated from the twisting power in the BPIII induces the axial chirality of the mesogenic core of **RM257**. This might be the origin of the memory effect. As the photopolymerization progresses, the induced chirality propagates to form a homochiral polymer network.

In order to understand why the polymer network can transform an achiral nematic LC into a chiral BPIII, we performed photopolymerization of achiral monomers with the BPIII polymer network as a template and investigated the chirality transfer from the BPIII polymer network to a helical polymer film [44]. We prepared a mixture of **AHCB** (14 wt%), **TMPTA** (81 wt%) and a photoinitiator (5.0 wt%) (Figure 36) and used the polymer network derived from the BPIII mixture containing **R-811** as a template. Figure 37 shows the CD spectra of the polymer network derived from the defects of the BPIII and the polymer film obtained by the polymerization with the polymer network as a template. They show negative signals; therefore, their optical senses are the same. Bisignate CD signals originated from the twisted conformation of the biphenyl unit of **AHCB** were not observed. This suggests that conventional achiral monomers without a chiral origin form a helical polymer by the photopolymerization with the polymer network as a template. According to the FE-SEM image of the BPIII polymer network, it has a porous structure and its pore sizes are about 100–200 nm. It seems to be possible that the reactive monomers gather in the pores, and they are photopolymerized along the cores. We surmise that the supramolecular chirality of the BP polymer network is transformed into the corresponding polymer film during the photopolymerization without inducing a microscopic molecular twist. We apply this hypothesis to the transformation of a nematic LC into BPIII as follows. The BPIII network memorizes the chirality of BPIII at a molecular level and transfers the chirality to conventional liquid-crystalline molecules possessing no chiral origin to form the helical assembly. Generally, rod-like LC molecules in a chiral field prefer to form a double-twist structure than a single twist structure; however, fitting these double-twist structures into the three-dimensional space comes with unfavorable defects. In the existence of the polymer network that is derived from the defects of the BPIII, the molecules form the more stable double-twist structure to produce a spaghetti-like tangled arrangement of DTCs.

## 6. Concluding Remarks

The structure of BPIII remained unclear for a long time. According to the simulation studies on BPIII and the SEM measurements of the polymer network derived from BPIII, we have clarified the structure of BPIII as follows. BPIII is a spaghetti-like tangled arrangement of DTCs with characteristic dynamics. The defect structure in BPIII is disorder with fluidity; therefore, the DTCs easily rearrange in a liquid-like fashion. Based on structure–property relationships in BPIII materials and the clarified BPIII structure, we have obtained a molecular design concept wherein molecular biaxiality and flexibility play an important role in stabilizing BPIII. Furthermore, we have proposed a ternary system for designing practical BPIII materials. This consists of a rod-like nematic mixture, a BPIII stabilizer, and a chiral dopant. Physical properties are adjusted by the formation of the nematic mixture. The BP stabilizer has biaxiality to stabilize DTCs and flexibility to induce disorder in the system. A rod-like nematic LC mixture doped with the BPIII stabilizer might be a host LC, which is converted to BPIII LC by adding a chiral dopant with high twisting power. BPIII consisting of the ternary system exhibited submillisecond response, high contrast, and wide viewing angle at room temperature without surface treatment or an optical compensation film. It is free from both hysteresis and residual transmittance, due to BPIII having no original state due to characteristic dynamics. High driving voltage remains an issue, but this can be solved by material and device developments. Interesting memory effects of the BP polymer networks reported by Coles et al. [47] and Chien et al. [49] can be explained as follows. The chiral field originating from the twisting power in BPIII induces the axial chirality of the mesogenic core of reactive monomer **RM257** existing in the defects. The BPIII polymer network memorizing the chirality of BPIII at a molecular level transfers the chirality to conventional liquid-crystalline molecules possessing no chiral origin to form DTCs in the presence of the network. The memory mechanism provides useful information for using BPs as a template to produce a porous polymer [12]. Amorphous BPIII with the same symmetry as an isotropic liquid is not of only fundamental interest but also presents superior potential for use in next-generation displays.

## Figures and Tables

**Figure 1 materials-17-01291-f001:**
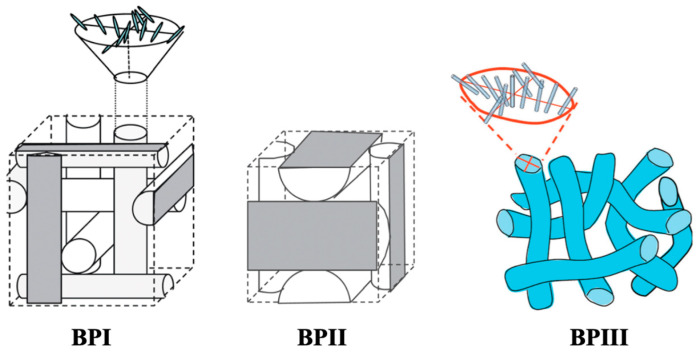
Schematic representations of the blue phase structures. The pictures of BPI and BPII are taken from [8]. Copyright (2013) Royal Society of Chemistry. The picture of BPIII is taken from [14]. Copyright (2016) Royal Society of Chemistry.

**Figure 2 materials-17-01291-f002:**
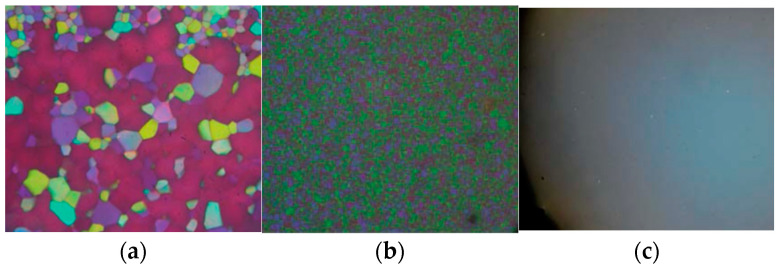
Photomicrographs of (**a**) BPI, (**b**) BPII, and (**c**) BPIII. These photographs were taken in our laboratory.

**Figure 3 materials-17-01291-f003:**
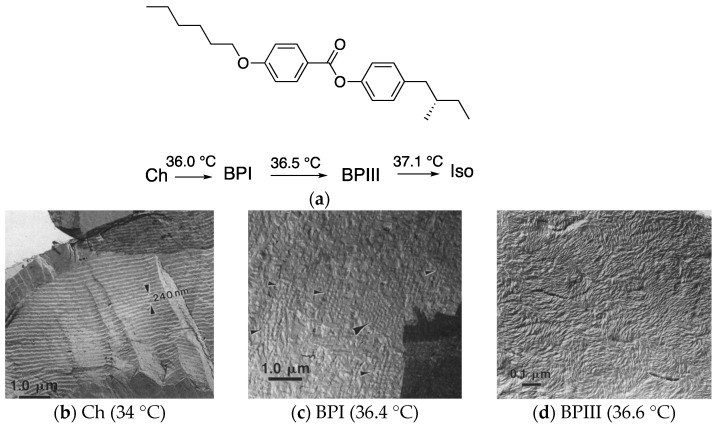
(**a**) Molecular structure and transition temperatures of **CE4**. The authors referred that **CE4** is a British Drug House designation for chiral 4-(2-methylpentylphenyl)-4′-hexyloxybenzoate, though **CE4** is generally used as the abbreviation for chiral 4-(2-methylbutyl)phenyl 4-hexyloxybenzoate. We show here the molecular structure of (S)- 4-(2-methylbutyl)phenyl 4-hexyloxybenzoate. (**b**) TEM image of a freeze-fracture replica of **CE4** quenched from the Ch phase at 34 °C to –190 °C. (**c**) That quenched from the BPI at 36.4 °C. (**d**) That quenched from the BPIII at 36.6 °C. Reprinted with permission from [45]. Copyright (1996) American Physical Society.

**Figure 4 materials-17-01291-f004:**
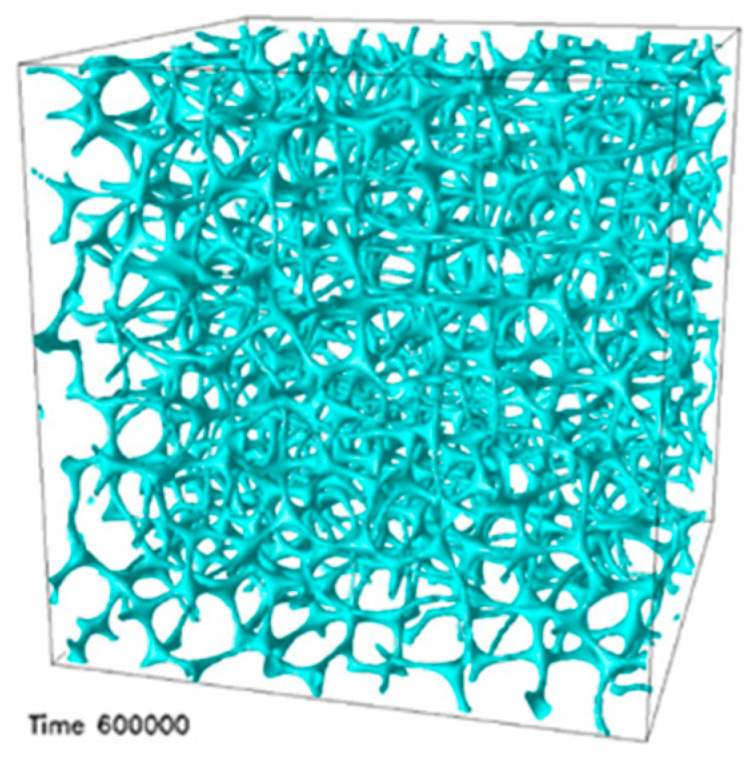
End-state disclination network in BPIII. Reprinted with permission from [23]. Copyright (2011) American Physical Society.

**Figure 5 materials-17-01291-f005:**
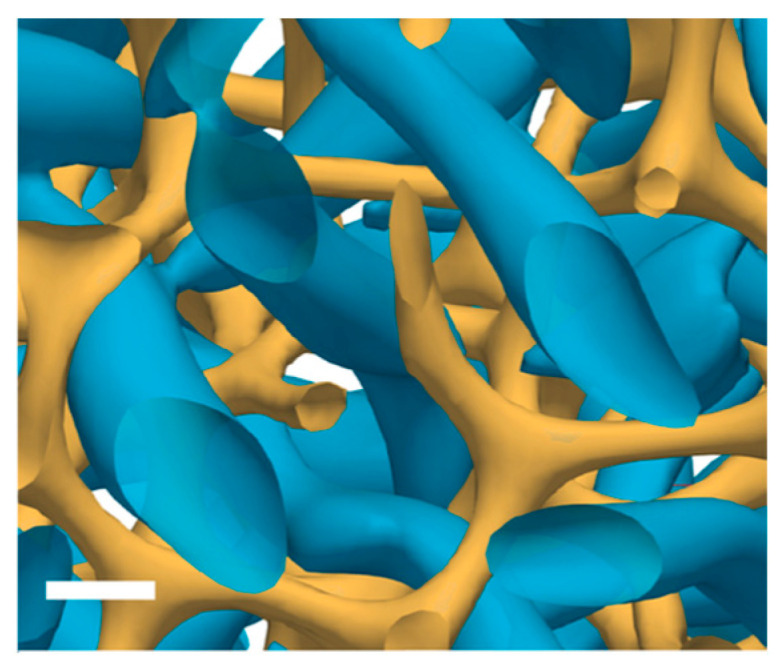
Numerical simulations of bulk BPIII. Bulk BPIII is a topological fluid of skyrmionic filaments (blue), enmeshed with a 3D network of –1/2 chiral singular lines (yellow). The scale bar represents 50 nm. This figure is taken from [46].

**Figure 6 materials-17-01291-f006:**
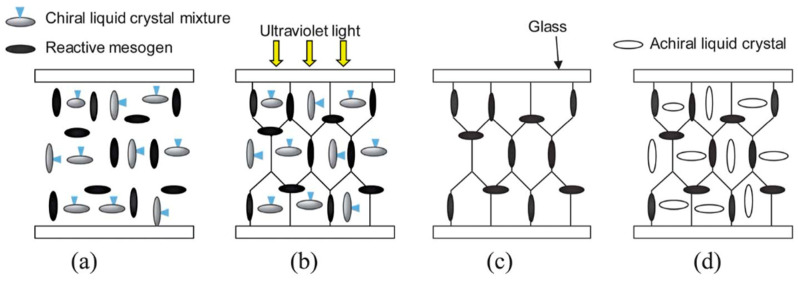
Schematic diagram of the formation of the 3D nanostructured polymer network [47]. (**a**) Chiral LC mixture forms a blue phase. (**b**) Cell is exposed to UV light to photopolymerize the reactive mesogens. (**c**) Removing the liquid crystal, chiral dopant, and remaining reactive mesogen–photoinitiator mixture creates a porous cast. (**d**) An unopened cell is refilled with the achiral nematic liquid crystal. The picture is taken from [9]. Copyright (2013) Royal Society of Chemistry.

**Figure 7 materials-17-01291-f007:**
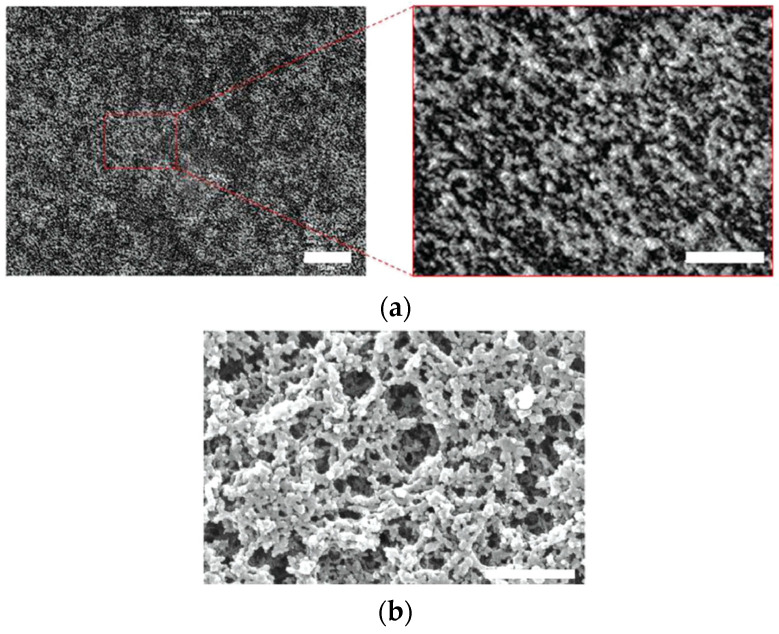
(**a**) CLSM image of a scaffold nanoengineered from BPIII. The scale bars are 15 µm for the complete image and 3 µm for the enlarged region. (**b**) SEM image of a scaffold nanoengineered from BPIII. The scale bar is 500 nm. Reprinted with permission from [49]. Copyright (2016) John Wiley and Sons.

**Figure 8 materials-17-01291-f008:**
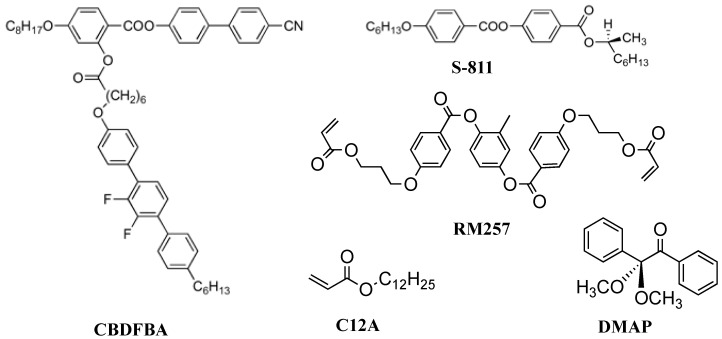
Molecular structures of the BP mixture.

**Figure 9 materials-17-01291-f009:**
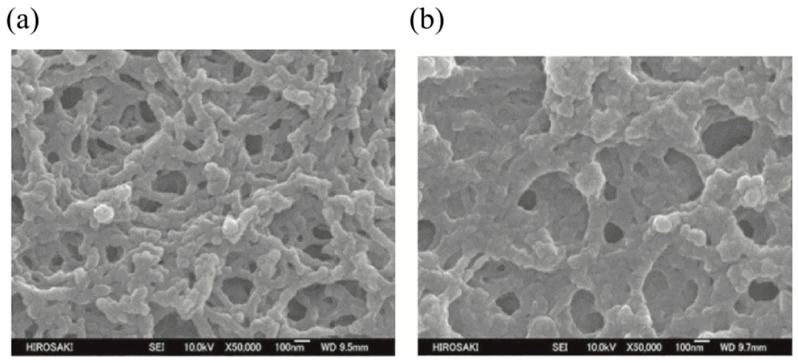
(**a**) FE-SEM image of the polymer network derived from BPII and (**b**) that derived from BPIII. The photographs are taken from [50]. Copyright (2023) Taylor & Francis.

**Figure 10 materials-17-01291-f010:**
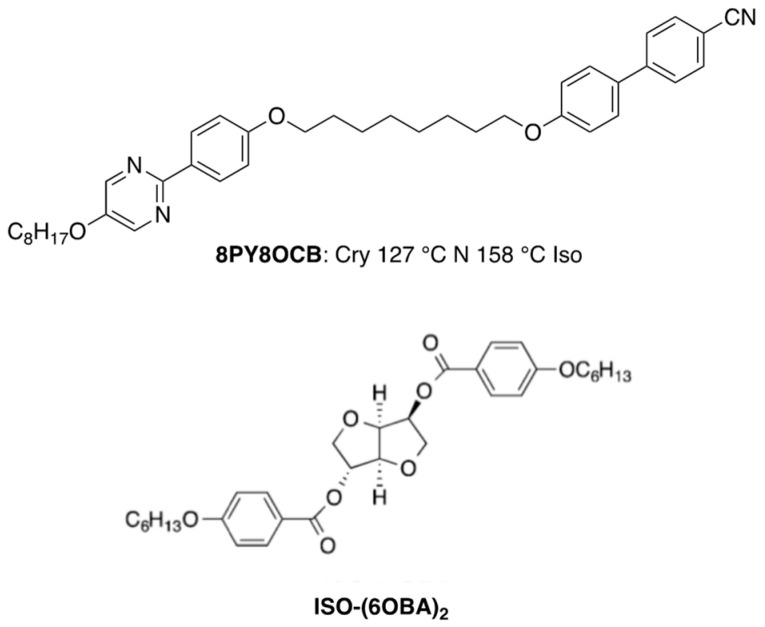
Molecular structures of **8PY8OCB** and **ISO-(6OBA)_2_**.

**Figure 11 materials-17-01291-f011:**
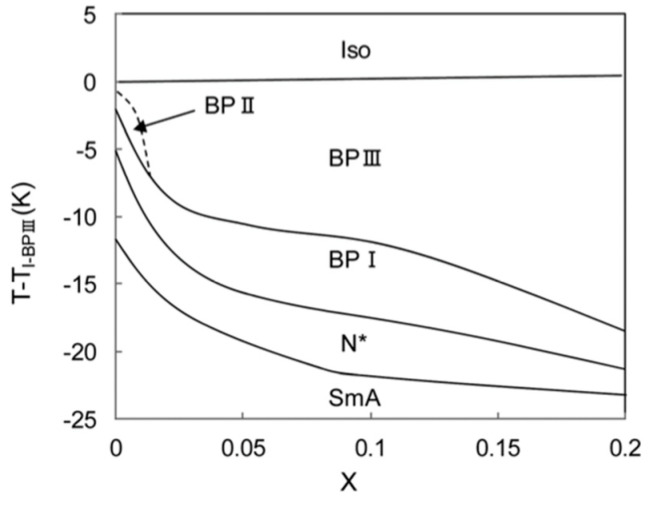
T–X phase diagram for a mixture of **CE8** and CdSe obtained in cooling runs. Reprinted with permission from [57]. Copyright (2010) American Physical Society.

**Figure 12 materials-17-01291-f012:**
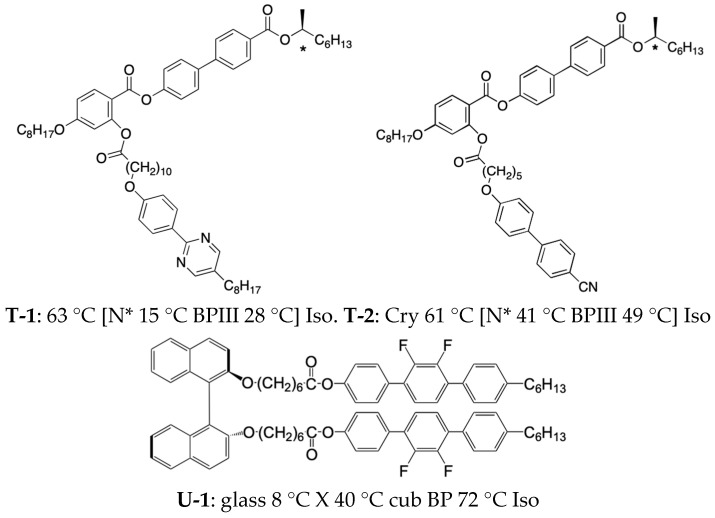
Molecular structures and phase transition temperatures of **T-1**, **T-2**, and **U-1**. The temperatures of **U-1** were observed on cooling. X is an unidentified smectic phase.

**Figure 13 materials-17-01291-f013:**
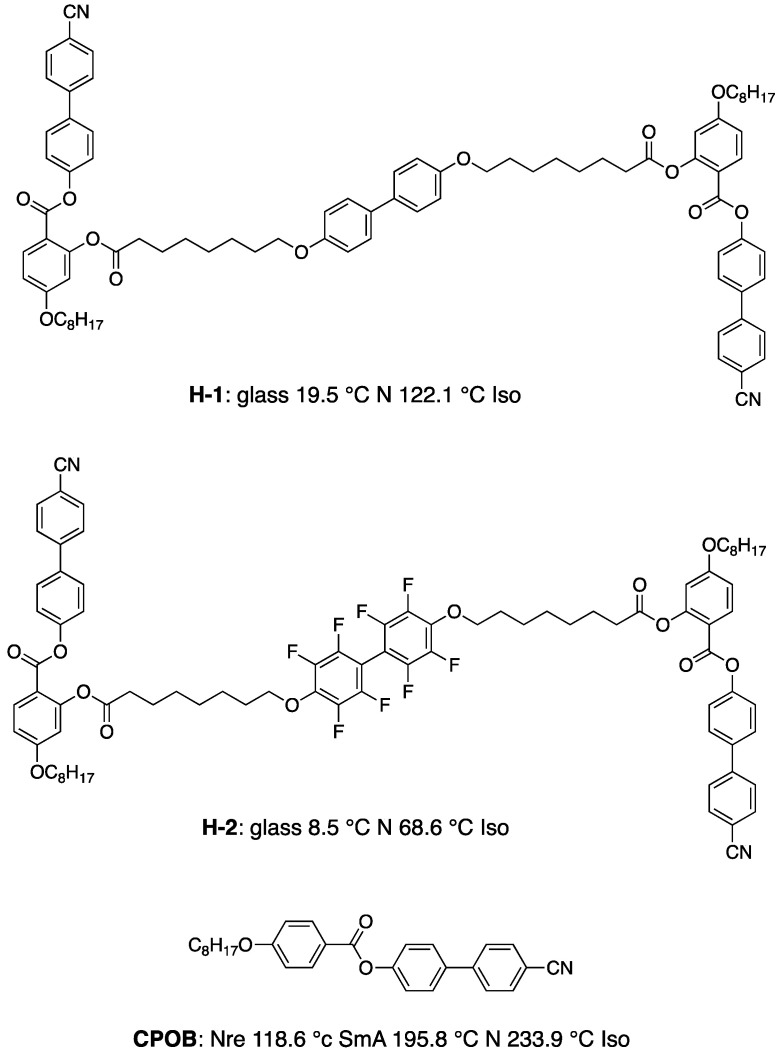
Molecular structures and phase transition temperatures on cooling of **H-1**, **H-2**, and **CPOB**.

**Figure 14 materials-17-01291-f014:**
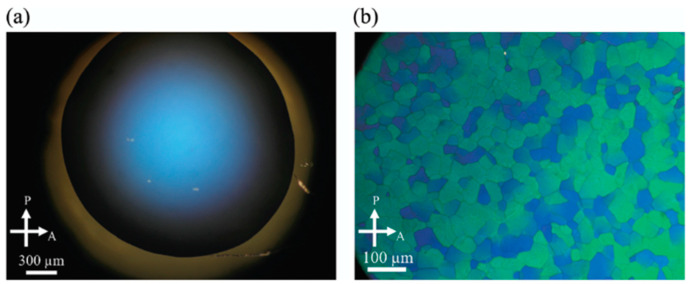
Optical textures of (**a**) BPIII of Mixture B at 23.4 °C and (**b**) cubic BP of Mixture C at 196.7 °C under crossed polarizers. The textures are taken from [61]. Copyright (2019) Royal Society of Chemistry.

**Figure 15 materials-17-01291-f015:**
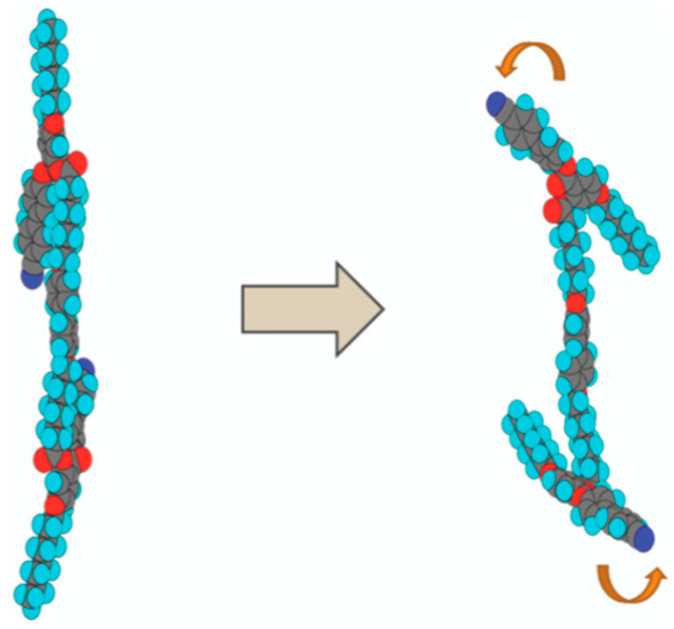
Model for the induced transiently twisted conformation of compound **H-1**. The picture is taken from [61]. Copyright (2019) Royal Society of Chemistry.

**Figure 16 materials-17-01291-f016:**
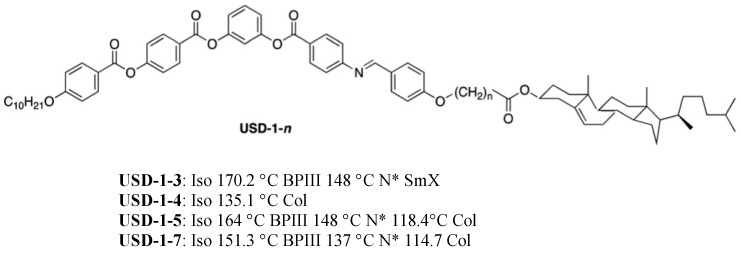
Molecular structure and phase transition temperatures (°C) of **USD-1-*n***.

**Figure 17 materials-17-01291-f017:**
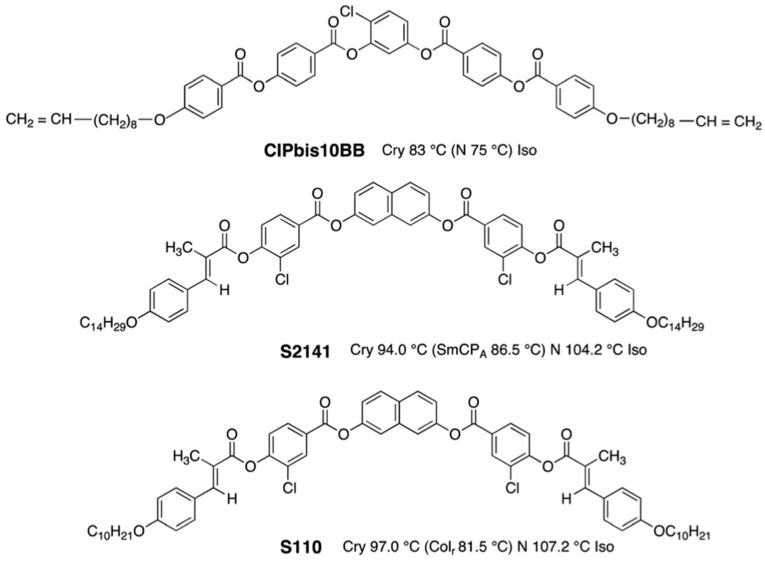
Molecular structures and phase transition temperatures of the bent-core molecules.

**Figure 18 materials-17-01291-f018:**
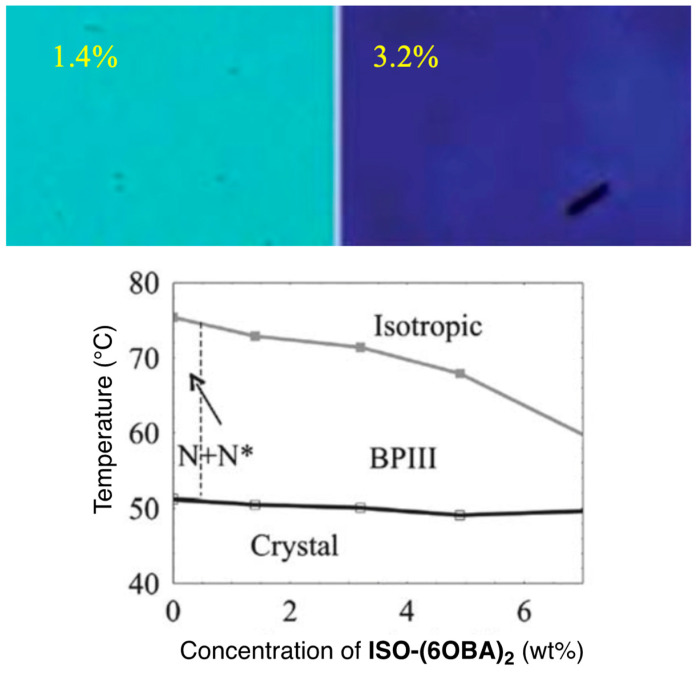
Phase diagram of **ClPbis10BB** and **ISO-(6OBA)_2_** on cooling. Optical textures of the BCN LC doped with the chiral compound (1.4 wt% and 3.2 wt%) are inserted. Reprinted with permission from [63]. Copyright (2010) Royal Society of Chemistry.

**Figure 19 materials-17-01291-f019:**
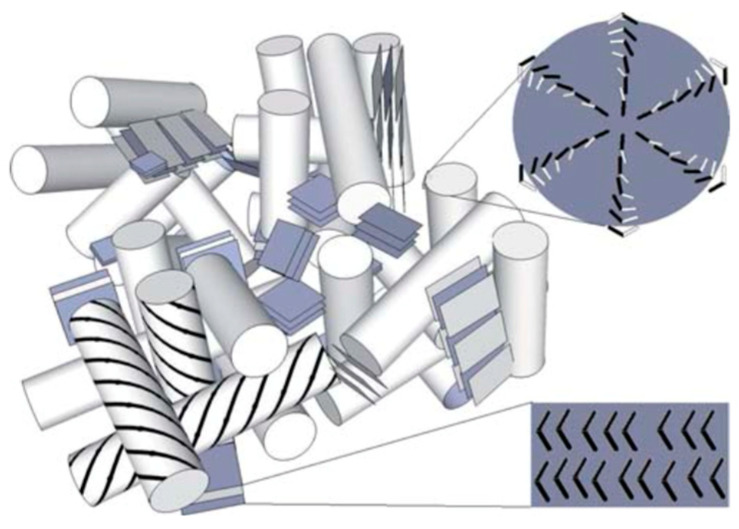
Illustration of the smectic cluster-stabilized BPIII structure of BCN LCs. Reprinted with permission from [63]. Copyright (2010) Royal Society of Chemistry.

**Figure 20 materials-17-01291-f020:**
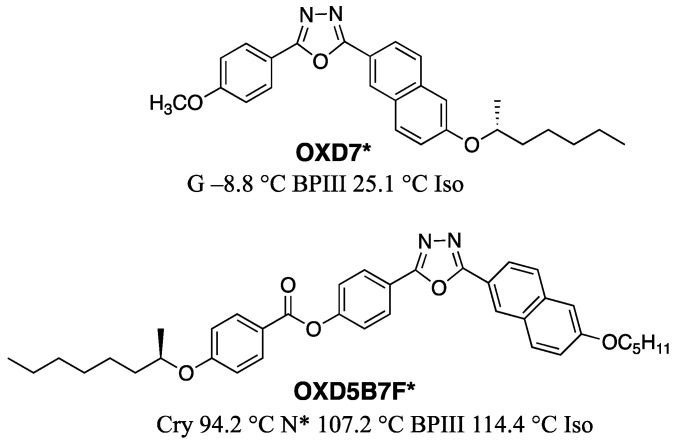
Molecular structures and cooling phase transition temperatures **OXD7*** and **OXD5B7F***.

**Figure 21 materials-17-01291-f021:**
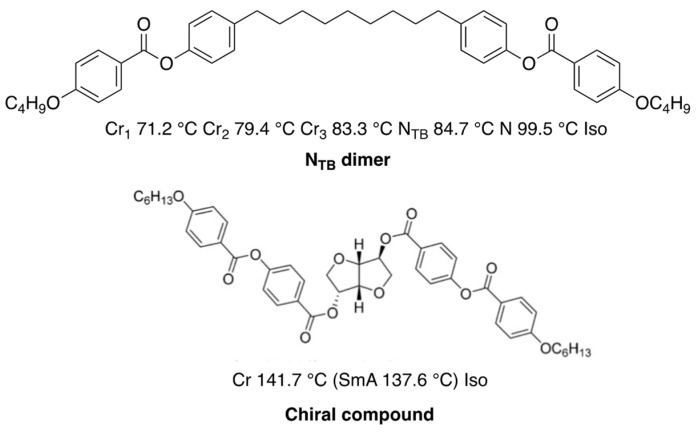
Molecular structures and phase transition temperatures of the N_TB_ dimer and the chiral compound.

**Figure 22 materials-17-01291-f022:**
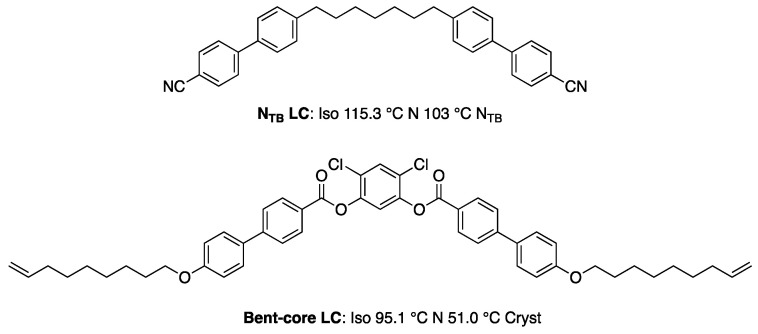
Molecular structures and phase transition temperatures of **N_TB_ LC** and **Bent-core LC**.

**Figure 23 materials-17-01291-f023:**
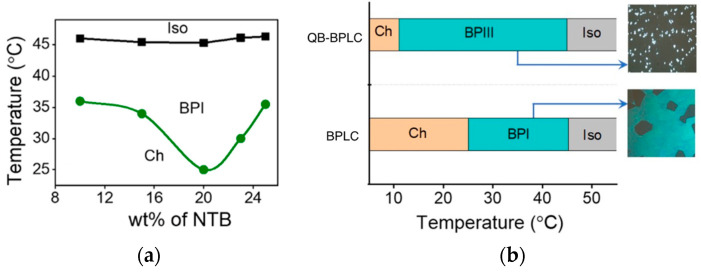
(**a**) Dependence of **N_TB_ LC** concentration on the thermal range of cubic BPI. The concentration of **S-811** was kept constant at 45 wt%. (**b**) The phase sequence of the BPLC and QD-BPLC mixtures. The BPLC mixture consists of 45 wt% of **S-811**, 20 wt% of **N_TB_ LC**, and 35 wt% of a commercial nematic mixture. The QB-BPLC mixture is a BPLC mixture doped with 0.005 wt% of QDs. Reprinted with permission from [68]. Copyright (2023) Royal Society of Chemistry.

**Figure 24 materials-17-01291-f024:**
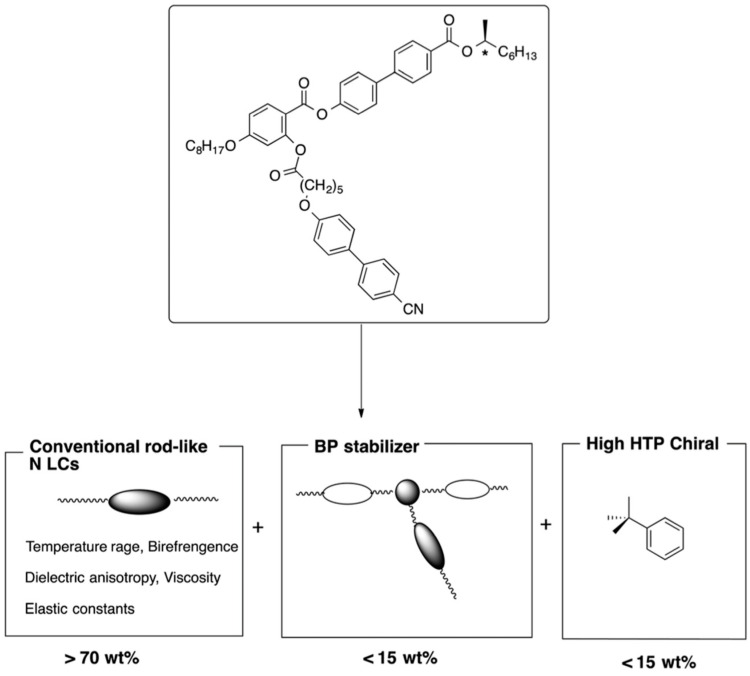
Design concept of the mixing system for a practical BPIII material. The figure is taken from [9]. Copyright (2013) Royal Society of Chemistry.

**Figure 25 materials-17-01291-f025:**
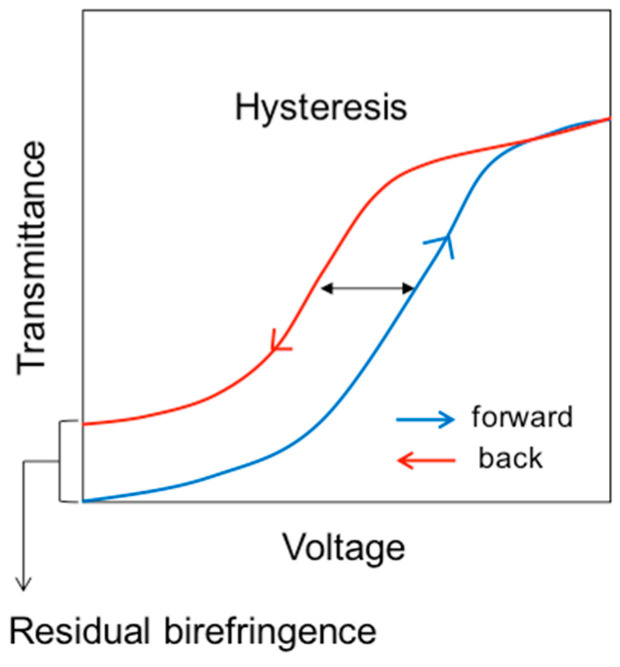
Voltage-dependent transmittance. The figure is taken from [9]. Copyright (2013) Royal Society of Chemistry.

**Figure 26 materials-17-01291-f026:**
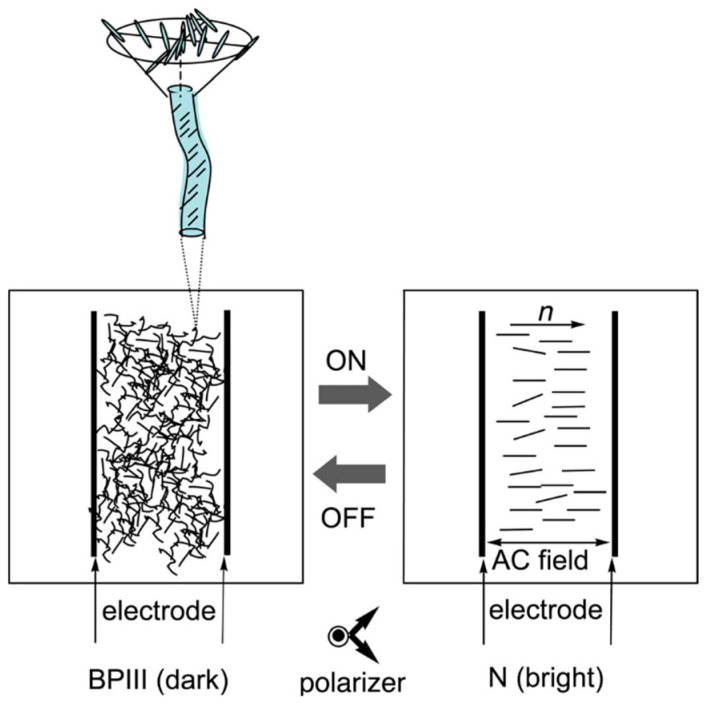
Schematic representation of the electric-field-induced phase transition between BPIII and N. Reprinted with permission from [74]. Copyright (2008) Society for Information Display.

**Figure 27 materials-17-01291-f027:**
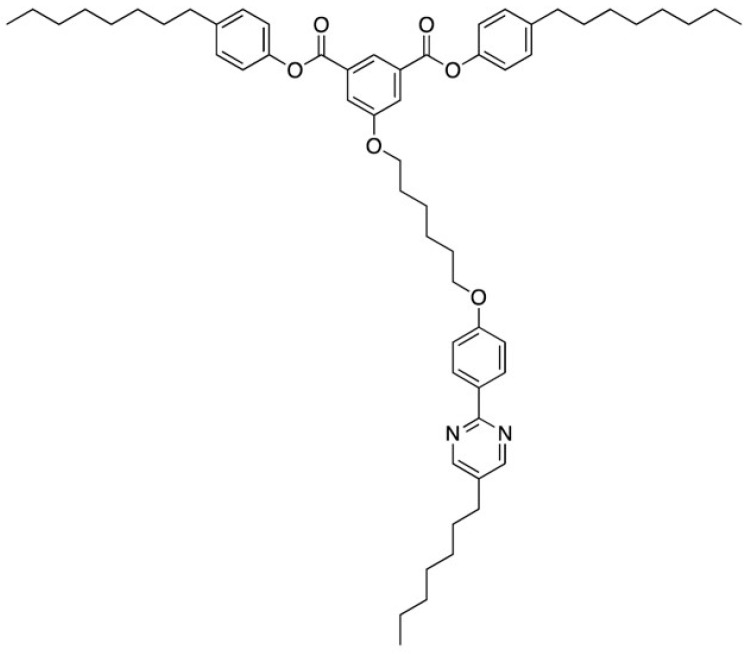
Molecular structure of the BP stabilizer.

**Figure 28 materials-17-01291-f028:**
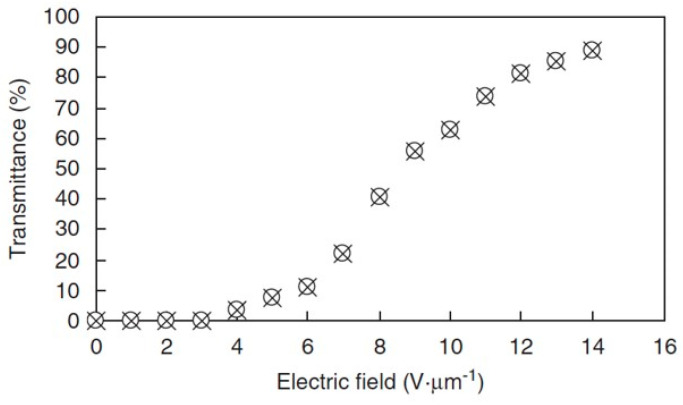
Optical transmittance of the chiral mixture as a function of an AC field at a frequency of 50 Hz in BPIII at 26 °C. The cell gap was 5 µm. Transmittance with 0% was calibrated by that of the cell under crossed polarizers. Transmittance with 100% was calibrated by that of the cell under parallel polarizers. Open circles and crosses denote ascending and descending processes, respectively. This figure is taken from [38]. Copyright (2011) The Japan Society of Applied Physics.

**Figure 29 materials-17-01291-f029:**
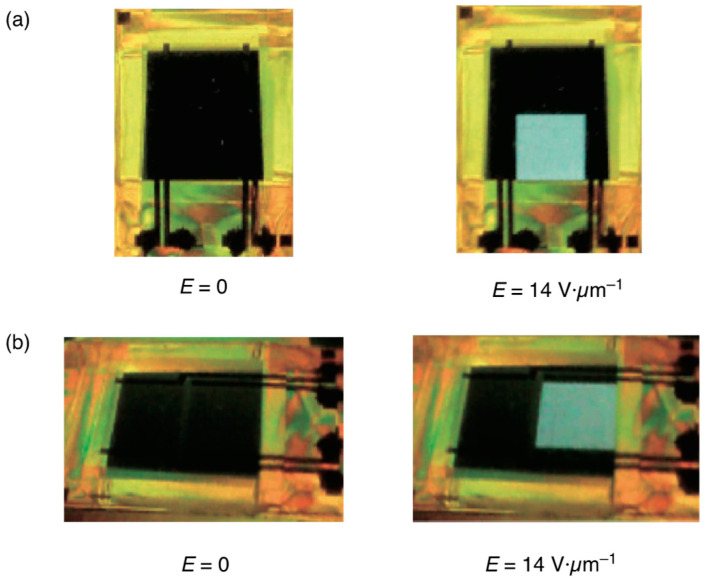
Viewing-angle dependence of transmittance in the BPIII cell. (**a**) Front and (**b**) oblique views with a viewing angle of approximately 45°. The photographs are taken from [38]. Copyright (2011) The Japan Society of Applied Physics.

**Figure 30 materials-17-01291-f030:**
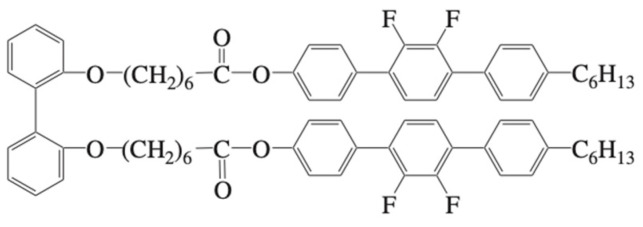
Molecular structure of the U-shaped BP stabilizer.

**Figure 31 materials-17-01291-f031:**
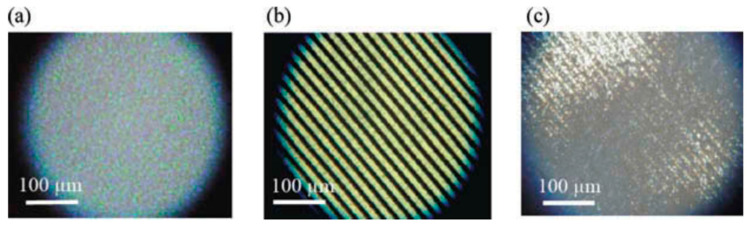
Optical texture of a sample in the PS-cubic BP at 25 °C without an electric field in the forward process (**a**), that with an electric field of 14 V·µm^−1^ (**b**), and that without an electric field in the backward process (**c**). The photographs are taken from [55]. Copyright (2015) Taylor & Francis.

**Figure 32 materials-17-01291-f032:**
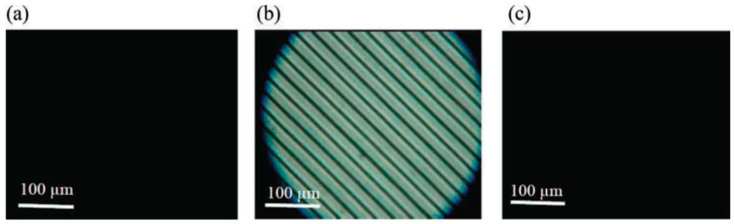
Optical texture of a sample in the PS-BP III at 25 °C without an electric field in the forward process (**a**), that with an electric field of 14 V·µm^−1^ (**b**) and that without an electric field in the backward process (**c**). The photographs are taken from [55]. Copyright (2015) Taylor & Francis.

**Figure 33 materials-17-01291-f033:**
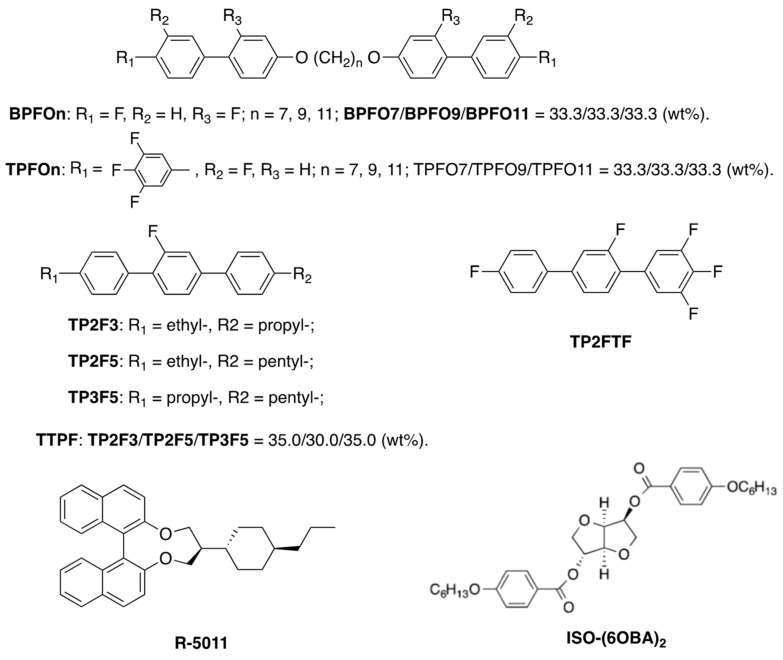
The molecular structures and materials.

**Figure 34 materials-17-01291-f034:**
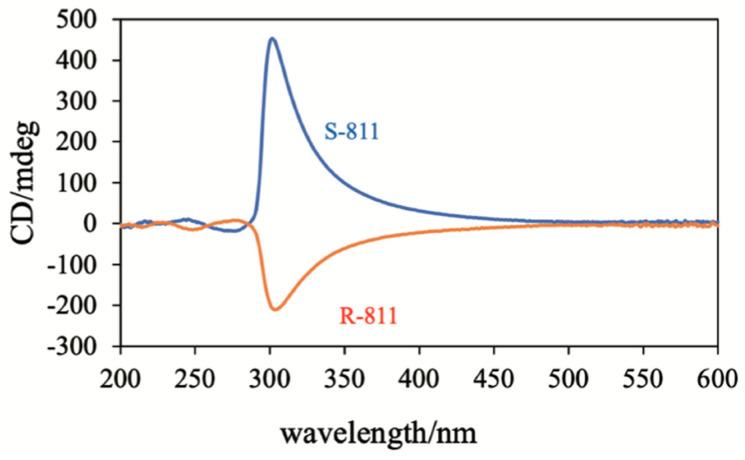
CD spectra of the BPIII polymer network derived from the BPIII mixture containing **S-811**(blue) and **R-811** (red). The figure is taken from [50]. Copyright (2023) Taylor & Francis.

**Figure 35 materials-17-01291-f035:**
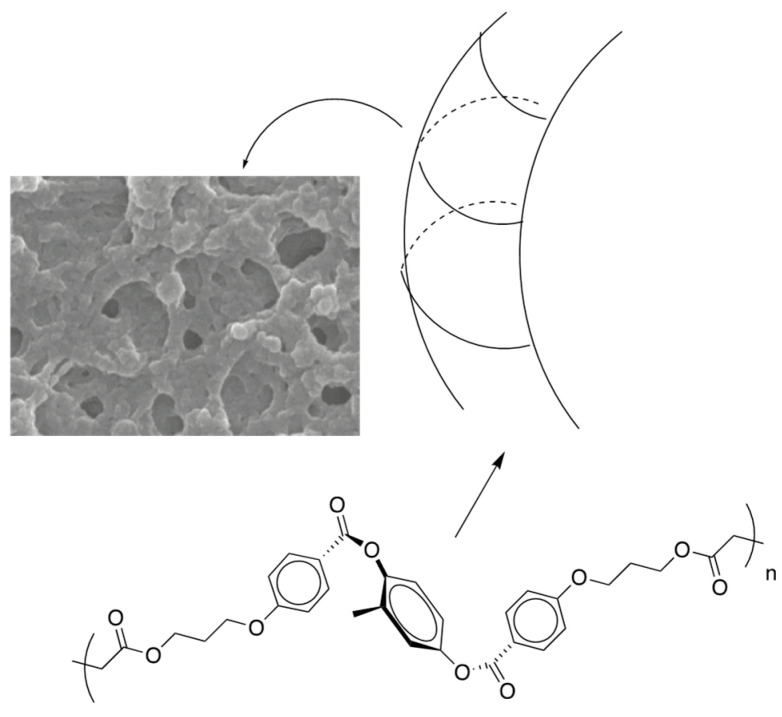
Schematic image of the formation of the BPIII network. The figure is taken from [50]. Copyright (2023) Taylor & Francis.

**Figure 36 materials-17-01291-f036:**
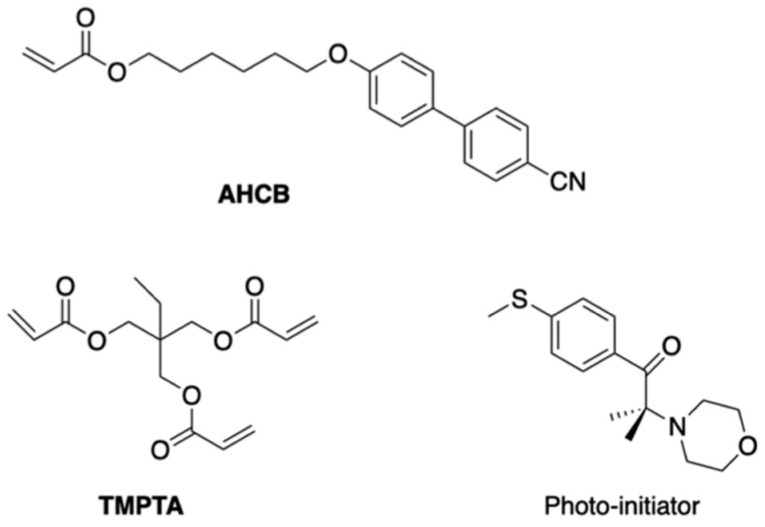
Molecular structures of **AHCB**, **TMPTA**, and a photoinitiator.

**Figure 37 materials-17-01291-f037:**
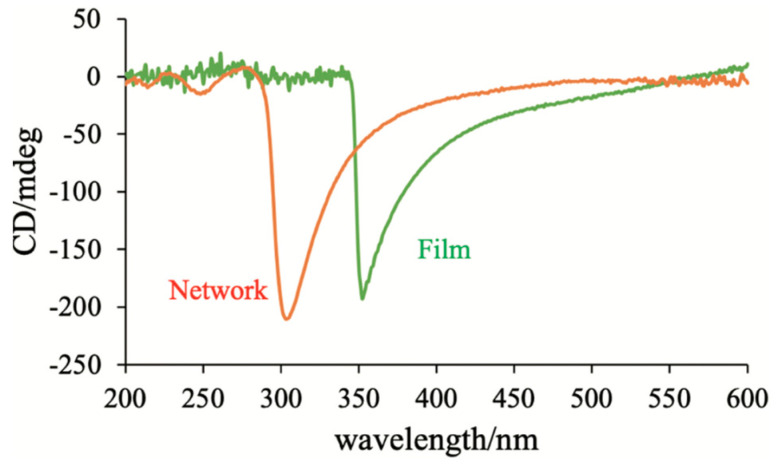
CD spectrum of the polymer network derived from the defects of the BPIII (red) and that of the polymer film obtained by polymerization with the polymer network as a template (green). Reprinted with permission from [50]. Copyright (2023) Taylor & Francis.

## Data Availability

All of the data and methodologies associated with the research described are available through the references.

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
