# Peer review of "Amorphous Blue Phase III: Structure, Materials, and Properties"

_materials, 2024, doi:10.3390/ma17061291_

Round 1

Reviewer 1 Report

Comments and Suggestions for Authors

I believe that the presented manuscript is relevant to the field and will constitute a valuable review article on BPIII. The only element that was missing was the lack of mention of conoscopic figures created for specific BPs.

Author Response

I would like to thank the reviewer for the very favorable evaluation and the helpful suggestion.

Reviewer’s comment
The only element that was missing was the lack of mention of conoscopic figures created for specific BPs.

Author’s reply
I have revised the text as follows.
“The cubic BPs were actively investigated by several experimental methods, such as the Kossel diagram technique [33,34] and well understood.”

I have added the following papers to the reference list.
33. Cladis, P.E.; Garel, T.; Pieranski, P. Kossel diagrams show electric-field-induced cubic-tetragonal structural transition in frustrated liquid-crystal blue phases. Phys. Rev. Lett. 1986, 57, 2841–2844.
34. Jéròme, B.; Pieranski, P. Kossel diagrams of blue phases. Liq. Cryst. 1989, 5, 799–812.

Reviewer 2 Report

Comments and Suggestions for Authors

Thank you for providing a comprehensive review of BP III.

I have several isolated comments that may help the paper be more readable: 

What is the view of the pictures in figure 3?  ( side view or from top)

In figure 28 what was the value of "0"  relative to crossed polarizers ;  and the value of "100" relative to parallel polarizers ( I saw a value of 89% but I am not sure what was 100% )

Figure 32 refers to a PB-BP III  that saturates at 14V/micron.   I found it interesting that this was the same value as the material without the polymer stabilization.   What that expected ?

Line 703  has a time listed as 02 ms   ,  Should that be 0.2 ms ?

line 758  and figure 37    refer to a polymer network and a polymer film ,  but I am not sure I understand the difference 

Author Response

I would like to thank the reviewer for the favorable evaluation and the helpful comments. I have revised the manuscript according to the reviewer’s comments.

Reviewer’s comment

What is the view of the pictures in figure 3?  ( side view or from top)

Author’s reply
It is the side view. I have revised the description as follows
“Figure 3 shows the TEM images of a fracture surface of CE4 in the cholesteric, BPI and BPIII phases.”

Reviewer’s comment
In figure 28 what was the value of "0" relative to crossed polarizers ;  and the value of "100" relative to parallel polarizers ( I saw a value of 89% but I am not sure what was 100% )

Author’s reply
Transmittance with 0% was calibrated by that of the cell under crossed polarizers. Transmittance with 100% was calibrated by that of the cell under parallel polarizers. 
I have added the above descriptions to the figure caption.

Reviewer’s comment
Figure 32 refers to a PB-BP III that saturates at 14V/micron.   I found it interesting that this was the same value as the material without the polymer stabilization.   What that expected ?

Author’s reply
The material used for the PB-BPIII is different from that without the polymer stabilization. The BP stabilizer for the material without the polymer stabilization is the T-shaped molecule (Figure 27), whereas that with the polymer stabilization is the U-shaped molecule (Figure 30). Therefore, I am not sure that the same in saturated electric field between with and without the polymer stabilization is due to the characteristic of BPIII. Unfortunately, I do not have a suitable explanation for the electro-optical behavior.

Reviewer’s comment
Line 703 has a time listed as 02 ms, Should that be 0.2 ms ?

Author’s reply
I thank the reviewer very much for the helpful comment. This is my mistake. I have corrected the value in the text.

Reviewer’s comment
line 758 and figure 37 refer to a polymer network and a polymer film, but I am not sure I understand the difference 

Author’s reply
The polymer network is a polymer derived from the defects of the BPIII, on the other hand, the polymer film is a polymer obtained by the photopolymerization of achiral monomers with the polymer network as a template.

We have revised the text and the figure caption as follows.
Text: “Figure 37 shows the CD spectra of the BPIII polymer network derived from the defects of the BPIII and the polymer film obtained by the polymerization with the polymer network as a template.”
Caption: “CD spectrum of the BPIII polymer network derived from the defects of the BPIII (red) and that of the polymer film obtained by the polymerization with the polymer network as a template (green).”

Reviewer 3 Report

Comments and Suggestions for Authors

The paper is a comprehensive and detailed review of current state of research of the liquid crystal Blue Phase III: molecular spatial packing structure, molecular chemical structure, physical properties, electro-optic response. Blue Phase III can offer a polarisation independent tunable photonic crystal for various applications and also presents some fundamental interest. The paper is well organised and well written, all the required details and analysis are included, conclusions and statements are well supported. I believe it can be accepted for publication as it is.

A couple of minor corrections in the abstract could be done:

Line 8. “the cylinder packing” - the cylinders are not introduced yet

Line 13. “DTCs” – also not introduced yet.

Author Response

I would like to thank the reviewer for the very favorable evaluation and the helpful comments. I have revised the manuscript according to the reviewer’s comments.

Reviewer’s comments
A couple of minor corrections in the abstract could be done:
Line 8. “the cylinder packing” - the cylinders are not introduced yet
Line 13. “DTCs” – also not introduced yet.

Author’s reply
I have revised the abstract accordingly as follows.

Lines 8: They are classified into three categories: blue phase I (BPI), blue phase II (BPII), and blue phase III (BPIII).
Line 12-13: BPIII is a ‘spaghetti’ like tangled arrangement of double-twist cylinders with characteristic dynamics.

Reviewer 4 Report

Comments and Suggestions for Authors This is a review manuscript on the structure, materials, and properties of blue phase BPIII. The manuscript provides a detailed introduction to the structural characteristics, electro-optic response, and latest material research progress of BPIII. The defect structure in BPIII is disordered and easily rearranged in a liquid like manner, resulting in no electro-optical hysteresis. At the same time, it retains characteristics such as sub millisecond response speed and high contrast, making it a promising next-generation liquid crystal display. I recommend publishing this manuscript, however, before it is published, it should be well revised. The following are some issues that need to be noted: 1. This manuscript summarizes the research results related to BPIII, but compared to published articles, the review is not comprehensive enough and does not separately summarize the general characteristics or mechanisms of each work. The manuscript lacks clear classification of published works, and there is less content in the last section on the review of memory effects in liquid crystal polymer networks, and no section title is provided. 2. The author provided a detailed introduction to their research work, while the introduction to other related blue phase works was not specific enough, and most of the references were relatively outdated, with only 5 articles cited in the past five years. Although there have been few articles on Blue Phase III in recent decades, the latest research related to Blue Phase should also be briefly introduced for beginner readers (The research work in Figure 33 is not actually related to BPIII). 3.There have been many methods for stabilizing the blue phase in recent years, far beyond polymer stabilization and nanoparticle stabilization. These works should also be briefly introduced in the manuscript if necessary, such as: [1] Advanced Materials, 2009, 21 (20): 2050-2053. [2] Rsc Advances, 2015, 5(82): 67357-67364. 4. The quality and clarity of the images in the manuscript need to be improved, for example, the photo in Figure 18 should be enlarged, and the three images in Figure 2 should have the same size.

Author Response

I would like to thank the reviewer very much for the very important comments. I have revised the manuscript as follows.

Reviewer’s comments

  1. This manuscript summarizes the research results related to BPIII, but compared to published articles, the review is not comprehensive enough and does not separately summarize the general characteristics or mechanisms of each work. The manuscript lacks clear classification of published works, and there is less content in the last section on the review of memory effects in liquid crystal polymer networks, and no section title is provided.

Author’s reply

The highlight of this manuscript is to clarify the structure of BPIII based on the theoretical and experimental works in the recent years (2020, 2022, 2023). In relation to the revealed structure of BPIII, we discuss the molecular design and the electro-optical effects in BPIII. In order to clear the purpose of this manuscript, I have added the following descriptions to Introduction.

“We clarify the structure of BPIII based on the recent theoretical studies and our experimental results in this review. The understanding on BPIII structure gives useful information to discuss the structure-property relations in BPIII. We propose the rational molecular design for stabilizing BPIII and explain the characteristic electro-optical effects in related to the revealed structure of BPIII. Finally, we discuss the mechanism of the memory effect of a polymer network derived from the defects of the BPIII. The presented findings provide important information to create BPIII-based functional materials.”

We have provided the section title as follows.

“5. The memory effect of a polymer network derived from the defects of the BPIII”

Reviewer’s comments

  1. The author provided a detailed introduction to their research work, while the introduction to other related blue phase works was not specific enough, and most of the references were relatively outdated, with only 5 articles cited in the past five years. Although there have been few articles on Blue Phase III in recent decades, the latest research related to Blue Phase should also be briefly introduced for beginner readers (The research work in Figure 33 is not actually related to BPIII).

Author’s reply

We have added the following 5 articles published in the past five years.

  1. Bagchi, K.; Emeršič, T.; Martínez-González, J.; de Pablo, J. J.; Nealey P. F. Functional soft materials from blue phase liquid crystals. Adv. 2023, 9, 9393.
  2. Hu, W.; Sun, J.; Wang, Q.; Zhang, L.; Yuan, X.; Chen, F.; Li, K.; Miao, Z.; Yang, D.; Yu, H. Humidity- responsive blue phase liquid-crystalline film with reconfigurable and tailored visual signals. Funct. Mater. 2020, 30, 2004610.
  3. Yang, Y.; Kim, Y.-K.; Wang, X.; Tsuei, M.; Abbott, N. L. Structural and optical response of polymer-stabilized blue phase liquid crystal films to volatile organic compounds. Appl. Mater. Interfaces 2020, 12, 42099–42108.
  4. Yang, Y.; Wang, L.; Yang, H.; Li, Q. 3D Chiral photonic nanostructures based on blue-phase liquid crystals. Small Sci.2021, 1, 2100007.
  5. Cordoyiannis, G.; Lavrič, M.; Trček, M.; Tzitzios, V.; Lelidis, I.; Nounesis, G.; Daniel, M.; Kutnjak, Z. Quantum dot-driven stabilization of liquid-crystalline blue phases. Frontiers in Phys, 2020, 8, 315.

We have added the brief descriptions for beginner readers to Introduction as follows.

“Cubic blue phases with a three-dimentiomal structure have recently attracted much attention due to their stimulus responsiveness, i.e., electrical, photonic, mechanical, and chemical stimulus [12]. They can be used not only for electro-optical applications but also for chemical sensors. For example, cubic BPs that change their reflection wavelength in response to chemical stimulus can give the basis of calorimetric sensors. The BP-based calorimetric sensors to detect humidity [15] or toluene [16] have been reported. Furthermore, the BP structure has potentiality to perform as a template to produce a higher ordered structure. Blue phase liquid crystals are expected to create functional soft materials [12,17].”

With respect to the work in Figure 33, it is very important methodology and I believe that it can be applied to the development of practical BPIII materials.

Reviewer’s comments

3.There have been many methods for stabilizing the blue phase in recent years, far beyond polymer stabilization and nanoparticle stabilization. These works should also be briefly introduced in the manuscript if necessary, such as: [1] Advanced Materials, 2009, 21 (20): 2050-2053. [2] Rsc Advances, 2015, 5(82): 67357-67364

 Author’s reply

I have briefly introduced the suggested papers as follows in Introduction.

“In addition, hydrogen-bonded assemblies for stabilizing cubic BPs were reported [27,28].”

  1. He, W.; Pan, G.; Yang, Z.; Zhao, D.; Niu, G.; Huang, W.; Yuan, X.; Guo, J.; Cao, H.; Yang, H. Wide blue phase range in a hydrogen-bonded self-assembled complex of chiral fluoro-substituted benzoic acid and pyrimidine derivative. Adv. Mater. 2009, 21, 2050–2053.
  2. Wang, J.; Shi, Y.; Yang, K.; Wei, J.; Guo, J. Stabilization and optical switching of liquid crystal blue phase doped with azobenzene-based bent-shaped hydrogen-bonded assemblies. RSC Adv. 2015, 5, 67357–67364

Reviewer’s comments

  1. The quality and clarity of the images in the manuscript need to be improved, for example, the photo in Figure 18 should be enlarged, and the three images in Figure 2 should have the same size.

Author’s reply

The photo in Figure 18 has been enlarged and three images in Figure 2 have been adjusted to have the same size.